# Renal Involvement in Congenital Cytomegalovirus Infection: A Systematic Review

**DOI:** 10.3390/microorganisms9061304

**Published:** 2021-06-15

**Authors:** María Ríos-Barnés, Clàudia Fortuny, Ana Alarcón, Antoni Noguera-Julian

**Affiliations:** 1Malalties Infeccioses i Resposta Inflamatòria Sistèmica en Pediatria, Unitat d’Infeccions, Servei de Pediatria, Institut de Recerca Sant Joan de Déu, 08950 Barcelona, Spain; mrios@sjdhospitalbarcelona.org (M.R.-B.); cfortuny@sjdhospitalbarcelona.org (C.F.); 2Departament de Pediatria, Universitat de Barcelona, 08950 Barcelona, Spain; anaalarcon@sjdhospitalbarcelona.org; 3Centro de Investigación Biomédica en Red de Epidemiología y Salud Pública (CIBERESP), 28029 Madrid, Spain; 4Red de Investigación Translacional en Infectología Pediátrica (RITIP), 28046 Madrid, Spain; 5Cervell Neonatal, Servei de Neonatologia, Institut de Recerca Sant Joan de Déu, Hospital Sant Joan de Déu, 08950 Barcelona, Spain

**Keywords:** congenital infection, congenital nephrotic syndrome, cytomegalovirus, ganciclovir, inclusion bodies, renal tubules, valganciclovir, viruria

## Abstract

Background: Congenital cytomegalovirus (cCMV) infection is the most frequent mother-to-child transmitted infection worldwide and a prevalent cause of neonatal disease and long-term morbidity. The kidney is a target organ for CMV, which replicates in renal tubules and is excreted in large quantities in urine for years in children with cCMV infection. Nonetheless, kidney disease has rarely been reported in cCMV-infected patients. Objective: We aimed to describe the available data on renal involvement in patients with cCMV infection at the pathologic, functional, anatomical, and/or clinical levels. Methods: A systematic search was performed in the MEDLINE/PubMed, SCOPUS, and Cochrane databases. Studies describing any renal involvement in fetuses or neonates aged ≤3 weeks at diagnosis of microbiologically confirmed cCMV infection were eligible. Results: Twenty-four articles were included, with a very low level of evidence. Pathologic findings in autopsy studies universally described CMV typical inclusion bodies in tubular cells. No functional studies were identified. cCMV infection was not associated with an increased risk of kidney malformations. Congenital nephrotic syndrome was the most common clinical condition associated with cCMV, but a causal relationship cannot be established. Conclusions: Typical pathological features of cCMV infection are very common in renal tissue, but they do not seem to entail significant consequences at the anatomical or clinical levels.

## 1. Introduction

Human cytomegalovirus (CMV) is a member of the β *Herpesvirinae* subfamily. CMV is usually inoculated in the healthy host through exposure of the upper respiratory or genital tract to contaminated fluids such as urine, saliva, semen, cervical secretions, and human milk. After primary infection, CMV viremia leads to dissemination to a wide range of tissues in various organs, where the characteristic “owl’s eye” intranuclear inclusions can be seen [1]. Among others, the former include the intestinal mucosa, the salivary glands, the lungs, and the kidneys. A primary CMV infection is usually asymptomatic or only mildly symptomatic in the immunocompetent host, in whom it usually presents with a mononuclear syndrome, and it universally leads to the lifelong establishment of a latent infection [2]. Persistent viral replication is commonly established in the kidney epithelium, among other cell types. Actually, the shedding of viral particles in saliva and urine or in cervical secretions often persists for years after the primary infection [3].

Despite CMV being able to evade the host immune response via multiple mechanisms, CMV-associated disease rarely affects the immunocompetent host after primary infection. In contrast, patients with severe impairment of immune function are at risk of severe CMV-associated disease, both after primary infection and upon CMV reactivation or reinfection with a different viral strain. The former includes those affected with cellular immunodeficiencies, such as solid organ or bone marrow transplant recipients, and severely immunosuppressed HIV-infected patients [4,5]. Antibody-mediated immunity is also critical in controlling CMV infection, as demonstrated by the different incidence and severity of CMV-associated disease in previously CMV seropositive or seronegative transplant recipients [6,7,8].

In kidney transplant recipients, tissue-invasive CMV disease is frequent and may affect a wide range of organs, most often the gastrointestinal tract (i.e., enteritis or colitis) [6,7]. Glomerulopathy, thrombotic microangiopathy, interstitial nephritis, and acute or chronic graft rejection have also been reported and put the patient at risk of allograft failure and death [6,7]. The use of prophylactic antivirals has had a dramatic impact on the incidence and severity of CMV disease after kidney transplant [9]. In allogeneic bone marrow transplant recipients who are at risk of CMV disease, CMV-associated acute kidney injury has been reported, increasing early mortality and accelerating the progression to chronic kidney disease in some patients [8]. Both universal prophylaxis and pre-emptive antiviral therapy (upon the identification of asymptomatic CMV viremia) have demonstrated a survival benefit, as well in this group of patients [10]. Finally, in the pre-antiretroviral therapy era, CMV was a frequent contributor to kidney disease in HIV-infected patients [11].

Congenital CMV (cCMV) infection is the most frequent congenital infection worldwide, with prevalence rates ranging from 0.2–0.7% of live births in high-income regions and to 1–6% in low- and middle-income regions [12,13]. A fetal CMV infection is more likely after maternal primary infection, but it can also occur upon reinfection with a different CMV strain or upon viral reactivation in a previously immunized pregnant woman. The earlier in pregnancy the infection occurs, the higher the odds are of developing severe fetal manifestations [14,15,16]. The most commonly described pathologic placental findings due to cCMV infection include chronic or necrotizing villitis, sclerosis of the villous capillaries, and chorionic vessel thrombosis [17,18]. These changes impair the placental transport and can lead to intrauterine growth restriction and, ultimately, to stillbirths.

At birth, 90% of newborns with cCMV infections are asymptomatic. In symptomatic newborns, typical clinical manifestations of cCMV infections include growth restriction, microcephaly, hepatosplenomegaly, jaundice, thrombocytopenia and petechiae, hepatitis, unilateral or bilateral sensorineural hearing loss, and abnormal neuroimaging. Other less common presentations of cCMV infections at birth include seizures, pneumonitis, leukemoid reaction, and blueberry or muffin rashes [19,20,21,22]. Symptomatic cCMV infections at birth are further categorized as mild or severe, the latter including those patients with central nervous system involvement or life-threatening disease. Babies with severe disease should be offered treatment with oral valganciclovir [22]. About 5–15% of those born with asymptomatic cCMV infection, and 50% of children who are born with the symptomatic disease, will develop significant sequelae in infancy—most commonly sensorineural hearing loss but, also, neurological impairment or retinitis [23,24,25].

As mentioned above, CMV inclusions have been described in most epithelial cells, including the renal tubular epithelium [1], and the virus is excreted in urine in large quantities for years upon cCMV infection [26]. Nonetheless, kidney disease is not part of the clinical spectrum of cCMV infection, and renal function impairment has only very rarely been reported in neonates and infants with this condition. In this systematic review, we aimed to describe the available data on renal involvement in patients with cCMV infection at the pathologic, functional, anatomical, and/or clinical levels.

## 2. Materials and Methods

This systematic review was conducted according to Preferred Reporting Items for Systematic Review and Meta-Analysis (PRISMA) [27] and aimed to assess whether the natural history of cCMV infection includes renal involvement at the pathologic, functional, anatomical, and/or clinical levels. Research ethics board approval was not required for this study.

We did not identify any published systematic reviews on this topic in the MEDLINE/PubMed databases. We also searched the PROSPERO database and made sure that no other systematic review on the topic was in process by another research group. The research protocol was registered in PROSPERO (registration number: CRD42021249225).

### 2.1. Study Selection and Search Strategy

A systematic literature search of the MEDLINE/PubMed and SCOPUS databases was performed. The MEDLINE/PubMed search strategy was developed using the National Library of Medicine’s medical subject heading (MeSH) browser to identify MeSH indexed terms (https://www.ncbi.nlm.nih.gov/mesh/?term=, accessed on 7 March 2021). MeSH terms associated with “Cytomegalovirus infections” were coupled with “Kidney disease” in a search string including “All fields”. In addition, the search in SCOPUS was conducted using a combination of key terms that included (“vertical” OR “mother to child” OR “mother-to-child” OR congenit* OR perinatal*) AND (“cytomegalovirus” OR “cmv”) AND (“kidney” OR “renal” OR nephr*) filtered by the “Title, abstract and keywords” field. The search in the Cochrane library identified no relevant studies. The search was limited to articles written in English or Spanish and involving humans aged below 18 years of age at diagnosis of CMV infection. There were no restrictions on the types of study or time restrictions. Reference lists of all relevant publications were also searched by hand to further identify potentially eligible articles using the snowballing method.

The original search was conducted in March 2021 independently by two reviewers (M.R.-B. and A.N.-J.) Figure 1 lists the terms used in the search string, as well as the algorithm for inclusion/exclusion.

The specific inclusion criteria were as follows: (1) studies focusing on the natural history of cCMV infection with regards to the renal involvement at the pathologic, functional, anatomical, and/or clinical levels and (2) previously published diagnostic criteria for cCMV infection [22]: positive conventional viral culture, positive shell vial culture, or CMV DNA detection by means of molecular assays in amniotic fluid or in urine, blood, kidney, or other tissues within the first 3 weeks of life or retrospectively in neonatal dried blood spot samples. When microbiological methods were not described, most commonly in old articles published when the former were not yet available, demonstrations of characteristic CMV-associated cytopathologic findings (i.e., basophilic intranuclear or eosinophilic cytoplasmic inclusions) with or without CMV-specific immunohistochemical staining techniques in the tissues of the fetus or the neonate within the first 3 weeks of life were considered, as well as evidence of cCMV infection. Given the exploratory nature of this systematic review, neither the assessment of therapeutic interventions nor the measurement of specific outcomes was required for inclusion. We excluded nonoriginal research and review articles, articles describing post-natal CMV infection (as opposed to cCMV infection), articles describing CMV-associated renal involvement in older children or adults affected with immunosuppressive conditions, and those articles for which the full text was not available.

### 2.2. Data Extraction and Quality Assessment

After the literature search, duplicate citations were removed, and the remaining references were independently screened by two investigators (M.R.-B. and A.N.-J.) in two stages. Initially, the titles and abstracts of the identified studies were assessed based on the inclusion and exclusion criteria, and, secondly, full papers were retrieved and further assessed. In case of disagreements regarding inclusion of an article, the final decision was reached by team consensus. The authors of two eligible studies were contacted by e-mail for additional clarifications, but only one responded. Dr. Chakraborty confirmed that all neonates and infants with cCMV infection in his cohort study fulfilled the inclusion diagnostic criteria [28].

Data were recorded separately and organized into Tables 1–4. For each selected publication, the following variables were extracted, if available: authors, year of publication, origin and type of study; total number of patients that the study included, and total number of patients with kidney involvement and cCMV that were eligible (Table 1); gender and gestational age or age at presentation; details on renal involvement at the pathologic, functional, anatomical, and/or clinical levels; other fetal or neonatal signs or symptoms consistent with cCMV infection (including, but not limited to, intrauterine growth restriction, microcephaly, chorioretinitis, thrombocytopenia, petechiae, hepatomegaly, splenomegaly, hepatitis, abnormal brain imaging, and sensorineural hearing loss); antiviral use; and final outcomes.

We performed a systematic narrative synthesis of the findings from the included studies, structured around the type of study design and type of renal involvement that was assessed. Strength of evidence was determined by means of the Levels of Evidence of the Oxford Centre for Evidence-Based Medicine [29].

## 3. Results

The initial search in MEDLINE/PubMed and SCOPUS yielded a total of 671 and 435 articles, respectively (Figure 1). The lists of articles were imported into reference managing software (Mendeley Ltd., New York, NY, USA), and a total of 24 duplicates were removed. Overall, 601 articles were excluded based on the initial selection criteria. After the title and abstract review, 422 further articles were excluded. A full-text and bibliography review of the remaining 59 articles was completed, and 22 of them were selected for inclusion. Hand-searching through the references of the selected articles identified two additional studies [30,31]. Finally, 24 articles were included in this systematic review. The level of evidence of the selected studies was between 4 and 5 [29].

The articles included in this systematic review were conducted in the United States (*n* = 8), Asia (*n* = 7), Europe (*n* = 7), and Australia (*n* = 2) and were published between 1964 and 2020. They comprised 2 cohort studies, 4 cross-sectional studies, 4 case series, and 14 case reports (≤3 cases) and included, overall, 100 patients with cCMV infections in whom some kind of kidney involvement was described (Table 1). Gender was reported in 52 patients, 22 (42.3%) of whom were females. cCMV infection was confirmed by means of molecular methods in amniotic fluid (*n* = 33), blood or urine (*n* = 30), kidney or other tissues (*n* = 17), positive viral cultures (*n* = 11), characteristic CMV-associated cytopathologic findings (*n* = 17), and immunohistochemical CMV-specific staining techniques (*n* = 13). Note that more than one diagnostic method was reported for some patients.

Most of the articles included in this systematic review reported pathologic findings in kidney samples of fetal (*n* = 8) or neonatal/infant (*n* = 8) autopsies or kidney biopsies in infants (*n* = 4). Four articles focused on the clinical signs or symptoms consistent with renal disease in patients with cCMV infection and included details on renal impairment at the biochemical or functional level in some of the patients; the search strategy identified no articles aimed at the study of renal function in children with cCMV infection. Three further studies described the renal involvement in the ultrasound (US) scans of fetuses known to be infected with CMV.

**Table 1 microorganisms-09-01304-t001:** Summaries of the articles that were included in this systematic review.

Ref.	Year	Country	Type of Study	Sample of Study	Included Cases	Eligible Cases
[32]	1964	USA	Case series	Neonate/infant autopsy	9	1
[33]	1969	USA	Case report	Kidney biopsy	1	1
[34]	1974	Italy	Case report	Neonate/infant autopsy	1	1
[35]	1985	USA	Case report	Kidney biopsy	1	1
[36]	1986	Japan	Case report	Fetal autopsy	1	1
[37]	1986	USA	Case series	Neonate/infant autopsy	5	5
[38]	1989	USA	Case report	Neonate/infant autopsy	1	1
[39]	1993	Australia	Case report	Clinical features	1	1
[40]	1993	USA	Case series	Clinical features	50	5
[41]	1993	USA	Case report	Clinical features and kidney biopsy	1	1
[42]	1993	China	Case report	Neonate/infant autopsy	1	1
[43]	1994	UK	Case report	Neonate/infant autopsy	1	1
[44]	1999	Korea	Case report	Fetal autopsy	3	3
[45]	2002	Germany	Case report	Neonate/infant autopsy	1	1
[46]	2006	USA	Case report	Fetal autopsy	1	1
[31]	2008	Italy	Cohort study	Fetal autopsy and clinical features	154	4
[30]	2009	Italy	Cross-sectional study	Fetal autopsy	34	26
[47]	2011	Australia	Cross-sectional study	Fetal autopsy	130	9
[48]	2014	France	Cross-sectional study	Clinical features	69	3
[49]	2016	China	Cross-sectional study	Fetal autopsy	436	7
[50]	2017	India	Case report	Neonate/infant autopsy	1	1
[51]	2019	China	Case series	Clinical features and kidney biopsy	14	4
[52]	2019	Serbia	Case report	Fetal autopsy	1	1
[28]	2020	India	Cohort study	Clinical features	80	20
Total number of eligible cases						100

### 3.1. Kidney Involvement at the Pathologic Level

Data on the histopathological findings in the kidney tissues were reported for 67 patients, including fetal autopsies on stillbirths (*n* = 10) and terminations of pregnancy (*n* = 39) [30,31,36,44,46,47,49,52], autopsies on infants with cCMV infections that died in the first months of age (ranging from minutes of life to 5 months of age) [32,34,37,38,42,43,45,50], and kidney biopsies (*n* = 4) in cCMV-infected infants with suspected renal disease [33,35,51]. The main histopathological features that were observed are summarized in Table 2.

**Table 2 microorganisms-09-01304-t002:** Description of the histopathological kidney involvement in patients with cCMV infections.

Ref.	N, Gender	Age	Kidney Histopathological Findings	Other Findings Likely Related to cCMV Infection	Unexpected Findings
**FETAL AUTOPSIES (gestational age in weeks)**					
[36]	1, male	22 wk	Intranuclear inclusion bodies in the kidneys	Intranuclear inclusion bodies in liver, lungs, pancreas, thyroid gland and placenta, not in heart; ventriculomegaly and hepatomegaly	Hydrops fetalis
[44]	3, male	30 wk	Nuclear and cytoplasmic inclusion bodies in the kidneys, especially in cortical tubules, without inflammation	Inclusion bodies in thymus and lungs	NR
[46]	1, NR	19 wk	CMV inclusions in tubular epithelium; focal areas of dystrophic calcifications with CMV-like inclusions in urinary bladder No findings in unaffected kidney	CMV inclusions in pneumocytes, bile duct epithelium, and placental villi	Dysplastic kidney
[31]	3, NR	NR	Massive involvement of kidney; no further details given	Severe disseminated disease in all cases, including central nervous system involvement in 2	NR
[30]	26, NR	20–21 wk	Inclusion bodies in epithelial cells (tubules, Bowman’s capsule); endothelial cells were also involved, particularly in the cortex, with severe focal inflammation with necrosis (6/26), mild inflammation without necrosis (13/26) or without inflammation (7/26)	Inclusions in other tissues also detected in most patients (placenta, pancreas, lungs, liver, heart and brain)	NR
[47]	9, NR	NR	Inclusion bodies in tubular epithelial cells (in 9 out of 20 autopsies)	Inclusion bodies in liver (10/20) and placenta (7/20)	NR
[49]	7; 4 males/3 females	32 wk	Intranuclear and intracytoplasmic inclusion bodies; microcalcifications in the kidney of a single patient	Inclusion bodies described also in liver, lungs, and pancreas	NR
[52]	1, female	31 wk	CMV cell inclusions in tubule epithelial cells and the epithelium of the parietal layer of Bowman’s capsule; modest mononuclear inflammatory infiltrate	Ventriculomegaly, lissencephaly, and brain calcifications; CMV cell inclusions in brain, lung and liver	Concomitant *Herpes simplex* virus infection
**INFANT AUTOPSIES (age at death)**					
[32]	1, female	1 d	CMV cell eosinophilic inclusion bodies	Generalized cytomegalic inclusion disease	Omphalocele
[34]	1, male	At birth	Inclusion-bearing cells with large necrotic areas in glomeruli	SGA, necrotizing encephalitis with ventriculomegaly, bilateral pneumonia, hepatosplenomegaly Inclusion bodies in liver, lungs, and brain	Lack of tubular inclusions Bullous dermatitis
[37]	1, female	4 mo	CMV cell inclusions in tubular epithelial cells, interstitial lymphocytic infiltrates, and inclusions in the endothelial cells of glomeruli (>90%)	Petechial rash, hepatosplenomegaly and microcephaly; disseminated CMV inclusions	Hemosiderosis
	1, male	5 mo	CMV cell inclusions in tubular epithelial cells, interstitial lymphocytic infiltrates, and inclusions in the endothelial cells of glomeruli	Petechiae, hepatitis, ascites	Inclusions observed only in kidneys
	1, male	At birth	CMV cell inclusions in tubular epithelial cells; interstitium hematopoiesis	SGA and prematurity (36 wk GA at birth) Microcephaly, ascites, extramedullary hematopoiesis; CMV inclusions in lungs, pancreas, brain, and pituitary gland	Hepatic fibrosis, hypoplasia of lungs and gallbladder
	1, male	2 d	CMV cell inclusions in tubular epithelial cells	SGA	Meconium aspiration syndrome, tetralogy of Fallot; inclusions observed only in kidneys
	1, female	6 wk	CMV cell inclusions in tubular epithelial cells, interstitial lymphocytic/neutrophilic infiltrates	SGA, petechiae, hepatosplenomegaly, jaundice, thrombocytopenia, extramedullary hematopoiesis CMV inclusions in lungs, pancreas, thyroid, and colon	Corneal and lenticular opacities, and uveitis; acute necrotizing pneumonia
[38]	1, female	2 d	Hypoplastic kidneys with severe interstitial nephritis, tubular atrophy and fibrosis, and CMV inclusions in the tubular cells	SGA, microcephaly and ventriculomegaly, pneumonitis, hepatosplenomegaly, extramedullary hematopoiesis	Maternal alcohol use during gestation, 4(del p) sd cranio-facial: dysmorphism, midline cleft, equinovarus, aorta coarctation, double uterus, cervix and vagina, pulmonary hypoplasia
[42]	1, male	14 d	CMV eosinophilic inclusions	Prematurity (31 wk GA at birth) Hepatosplenomegaly, thrombocytopenia, jaundice, and brain calcifications; inclusion cells in the liver, spleen, and adrenal and thyroid glands	NR
[43]	1, male	3 mo	Intranuclear inclusion bodies in renal tubules	NR	NR
[44]	1, female	At birth	Inclusion bodies in the kidneys (distal > proximal tubules) with mild inflammation	Prematurity (24 wk GA at birth) Brain calcifications, ventriculomegaly, extramedullary hematopoiesis; inclusion bodies in brain, thymus, heart, liver, and lungs	Hydrops fetalis
	1, female	3 d	Inclusion bodies in proximal and distal tubular epithelial cells and in glomerular endothelial cells; moderate interstitial inflammation and foci of necrosis	SGA, thrombocytopenia, spleen hematopoiesis Inclusion bodies in lungs and liver; brain was not examined	Aorta coarctation, pulmonary hypertension, and congestive heart failure
[45]	1, NR	At birth	Intranuclear inclusion bodies predominant in kidney epithelial cells	Prematurity (32 wk GA at birth) Hepatosplenomegaly, jaundice, and petechial bleeding, extramedullary hematopoiesis Disseminated CMV disease which correlated with clinical and pathological findings	Hydrops fetalis
[50]	1, male	2 d	Intranuclear inclusions surrounded by clear halo along with cytoplasmic inclusions in tubular lining cells and vascular endothelium; lymphocytic infiltrate in interstitium	SGA and prematurity (34 wk GA at birth) CMV inclusions in lungs and liver	HIV-exposed baby
**INFANTS’ KIDNEY BIOPSIES (age)**					
[33]	1, male	2 d	Intranuclear inclusion bodies	Chorioretinitis (left eye), splenomegaly	Fused kidneys in horseshoe configuration; multi-cystic non-functioning left kidney
[35]	1, male	10 d	Nuclear and cytoplasmic inclusions in tubular epithelial cells Interstitial mononuclear and T-cell infiltration	Prematurity (36 wk GA at birth), ascites, hepatosplenomegaly, and thrombocytopenia Progressive neurological deterioration	NR
[41]	1, male	6 wk	Increased mesangial matrix and glomerular sclerosis; nuclear and cytoplasmic CMV inclusions in the tubular epithelia; patchy moderate lympho-mononuclear inflammatory cellular infiltrate	Prematurity (34 wk GA at birth) Hepatosplenomegaly, hepatitis and thrombocytopenia at birth; subependymal cyst in cerebral US	Long-term unresolved proteinuria and pathologic features consistent with genetic CNS
[51]	1, male	2 mo	Mild mesangial proliferative glomerulonephritis disease and tubular epithelia degeneration; no typical CMV inclusion bodies and negative CMV DNA tests in renal tissue	Transient response of proteinuria to treatment with ganciclovir	*NPHS1* (Finnish-type CNS) and *COL4A5* gene mutation (Alport syndrome); no signs or symptoms consistent with cCMV infection

Abbreviations: CNS, congenital nephrotic syndrome; d, day; GA, gestational age; HIV, human immunodeficiency virus; mo, month; NR, not reported; SGA, small for gestational age; US, ultrasound; wk, week.

Two Italian studies described the histological findings in the autopsies of fetuses with cCMV infections. Guerra et al. [31] reported the US findings of 650 pregnant women with primary CMV infections during a 10-year period. Overall, 154 fetuses were diagnosed with cCMV infections. The termination of pregnancy was performed in nine cases upon severe fetal malformations, and one patient died in utero at 22 weeks of gestational age. Severe involvement of the kidneys was reported in three out of 10 fetuses, together with disseminated CMV disease. No further details were given. Later on, Gabrielli et al. [30] reported a comprehensive histological study in 31 cCMV-infected fetuses. CMV inclusion bodies were found in the placentas (100%), pancreases (100%), lungs (87%), kidneys (87%), livers (71%), brains (55%), and hearts (44%). In renal tissue, the predominant target cells were epithelial cells in the tubules and Bowman’s capsule; endothelial cells in the renal cortex were less commonly affected. The degree of tissue inflammation (assessed by means of the leukocyte antigen CD45) correlated with the proportion of CMV-infected cells; severe inflammation was also associated with necrosis.

Other studies aimed to elucidate the role of cCMV infection in different pregnancy outcomes. Iwasenko et al. [47] collected the livers, kidneys, and placentas of 130 stillbirths of unknown etiology and identified CMV DNA in 20 cases (13%). The kidneys were affected in approximately half of these (nine out of 20). Of note, fetal thrombotic vasculopathy was the only histopathological abnormality associated with cCMV infection as compared with uninfected stillbirths. Lin et al. [49] identified CMV DNA in seven out of 436 (1.6%) fetuses with severe malformations and terminated pregnancy in a 7-year period in China. The kidneys were affected in all seven cases. Both studies [47,49] reported inclusion bodies in tubular epithelial cells as the main histopathologic finding in the renal tissue.

The rest of the articles describing the autopsies of cCMV-infected fetuses or neonates/infants and, also, those describing kidney biopsies in infants consisted of case series and case reports [31,32,33,34,35,36,37,38,42,43,44,46,49,50,51,52,53]. In the former, severely affected fetuses or neonates/infants were reported, with gross examination and histopathological findings consistently associated with cCMV infection. Among the infants who underwent kidney biopsies, two showed pathologic findings consistent with a congenital nephrotic syndrome (CNS) of genetic origin [41,51]. Some authors also reported unexpected findings in which a causal relationship with cCMV infection was difficult to establish, such as dysplastic or multi-cystic kidney [33,46], omphalocele [32], hemosiderosis [37], congenital heart disease [37,44], corneal opacities and uveitis [37], and bullous dermatitis [34]. Finally, an alternative confirmed diagnosis of the renal condition was reported in some cases [38,54], implying that cCMV infection was unlikely to have been associated with kidney involvement in those patients.

Overall, the intranuclear and intracytoplasmic inclusion bodies typical of CMV infection were the most commonly described histopathological findings in the renal tissue of the patients included in this systematic review (Table 3). In most cases, inclusion bodies were reported in the tubular epithelium, followed by the epithelium of the Bowman’s capsule. The inclusion bodies were rarely described in endothelium cells. Some degree of mostly lymphocytic interstitial inflammation in the renal tissue was also observed, which seldom was associated with the foci of necrosis. Unexpected histopathologic findings were described as well, including calcifications in the urinary bladder [46] and kidney [49], hematopoiesis in the renal interstitium [37], and necrotizing glomerulonephritis [34]; as mentioned above, changes consistent with the CNS of genetic origin (i.e., increased mesangial matrix and progressive glomerulosclerosis) were observed in the kidney biopsies of two infants with proteinuria [41,54], together with typical CMV tubular inclusion bodies in one of them. Inclusion bodies in several other tissues (placenta, liver, lungs, brain, pancreas, thymus, thyroid gland, and heart) were reported in most of the patients as well.

**Table 3 microorganisms-09-01304-t003:** Details of the most common histopathological findings in renal tissue ((++) main finding, (+) reported, and (-) not reported/unknown). Intranuclear/intracytoplasmic inclusion bodies were reported in all cases, but details on the cells that were affected were not always provided. Chen et al. [51] did not report typical CMV pathological findings.

Ref.	*N*	Intranuclear/Intracytoplasmic Inclusion Bodies				Interstitial Inflammation	Necrosis
		In Kidneys	In Tubular Epithelium	In Bowman’s Capsule	In Endothelial Cells		
[52]	1	+	++	++	-	-	-
[30]	26	+	++	+	-	++	+
[44]	3	+	++	+	+	+	+
[37]	5	+	++	+	+	+	+
[50]	1	+	+	-	+	+	-
[46]	1	+	++	-	-	-	-
[45]	1	+	++	-	-	-	-
[43]	1	+	+	-	-	-	-
[38]	1	+	+	-	-	+	-
[41]	1	+	+	-	-	+	-
[35]	1	+	-	-	-	+	-
[34]	1	+	-	-	+	-	+
[49]	7	+	-	-	-	-	-
[47]	9	+	-	-	-	-	-
[36]	1	+	-	-	-	-	-
[33]	1	+	-	-	-	-	-
[31]	3	+	-	-	-	-	-
[32]	1	+	-	-	-	-	-
[42]	1	+	-	-	-	-	-

### 3.2. Kidney Involvement at the Clinical Level

Three articles in this systematic review (one cohort study, one case series, and one case report) described the occurrence of CNS in patients with cCMV infection, overall including 25 infants (Table 4) [28,41,51]. CNS is characterized by heavy proteinuria, hypoalbuminemia, hyperlipidemia, and edema that present by the age of 3 months. CNS has a genetic basis in most of the cases, but secondary CNS has been described with toxins (i.e., mercury exposure); maternal systemic lupus erythematosus; and several perinatal infections such as CMV, syphilis, rubella, human immunodeficiency virus, hepatitis B virus, and toxoplasmosis [55,56]. A proper differential diagnosis between the genetic and secondary forms of CNS is critical, since the former do not respond to therapy and require kidney transplantation, while specific therapy in secondary causes of CNS may be curative in some patients [57].

Chatterje et al. [28] identified 20 neonates with cCMV infection and CNS (Group 1) and compared them to three control groups: 20 neonates with CNS but no cCMV infection (Group 2), 20 with cCMV infection but no CNS (Group 3), and 20 without CNS or cCMV infection. All patients were followed-up to 1 year of age. Many patients in Groups 1 and 3 presented signs or symptoms consistent with symptomatic cCMV infection, including thrombocytopenia (50%), hepatitis (40%), hepatosplenomegaly (52%), chorioretinitis (35%), and central nervous system impairment (40%). At inclusion, no differences in gender, age, gestational age at birth, weight or length at birth, and Apgar score were observed between the groups, but smaller head and chest circumferences were found in patients with cCMV infection (i.e., Groups 1 and 3), and larger abdomen circumferences were found in patients with CNS (i.e., Groups 1 and 2). As expected, the albumin, cholesterol, triglyceride levels, and proteinuria were significantly abnormal in patients with CNS (i.e., Groups 1 and 2) at inclusion; except for proteinuria, all the parameters remained significantly impaired at 12 months of age in both groups. Patients with CNS (i.e., Groups 1 and 2) often presented with hydronephrosis (70%), diffuse mesangial sclerosis (60%), or focal tubular atrophy (60%) without differences between those with and those without concomitant cCMV infection. Neonates with CNS and cCMV infection presented significantly higher levels of C-reactive protein, interleukin-6, and interleukin-1β at inclusion when compared with those with CNS only; these findings are consistent with active inflammation, which could potentially increase the glomerular damage [58]. Although the authors stated that patients in Group 1 were categorized according to the CNS symptom response to antivirals, no details on the specific antiviral treatment were given. Nevertheless, the albumin levels were higher and the cholesterol levels and proteinuria were lower in Group 1 patients compared to Groups 2 patients at the 12 months of age follow-up. In the phylogenetic analysis of the 20 viral strains from the patients in Group 1, 13 of the sequences formed a distinct cluster compared to the reference strains.

In a cross-sectional study, Chen et al. [51] identified 14 infants with CNS during a 23-year period in a single institution in China. In eight patients, concomitant CMV infection was diagnosed, but only four of those fulfilled the age criteria for cCMV infection (Table 4). None of them presented with the typical signs or symptoms of cCMV infection. Genetic testing identified *NPHS1* mutations in three of the patients and was not performed in the remaining one. *NPHS1* mutations are known to cause Finnish-type CNS, the second-most frequent type of CNS, which is inherited as an autosomal recessive trait [59]. In one of the infants, the gene that causes Alport syndrome was also identified (mutation *COL4A5*). His kidney biopsy showed mild mesangial proliferative glomerulonephritis disease and tubular epithelia degeneration but not typical cytomegalic inclusion bodies, and CMV DNA molecular tests in the renal tissue yielded negative results. Kidney biopsies were not performed in the rest of the children. All the patients were treated with intravenous ganciclovir uneventfully, but the treatment was curative only in one of the cases. Among the other children, two patients died in the first year of life, and one is still being monitored with persistent proteinuria.

**Table 4 microorganisms-09-01304-t004:** Clinical details of the infants with cCMV infection and congenital nephrotic syndrome.

Ref.	Gender, Age	Symptoms at Presentation	Ultrasound Findings	Genetic Study	Outcome
[41]	M, 21 d	Edema	Increased echogenicity of both kidneys	NP	Alive (age NR)
[54]	M, 2 d	Edema	Mild hydronephrosis	Finnish type	Death (10 mo)
	F, 20 d	Ascites	Normal	NP	Alive (9 y)
	M, 15 d	Asymptomatic proteinuria	Normal	Finnish type + Alport syndrome	Alive (13 mo)
	M, 19 d	Edema	Normal	Finnish type	Death (2 mo)
[28]	9M/11F, 20 d	NR	Hydronephrosis (75%)	NP	Alive (12 mo)

Abbreviations: d, day; F, female; M, male; mo, month; NP, not performed; NR, not reported; y, year.

In 1993, Batisky et al. [41] reported a 5-week-old male infant with generalized edema and severe proteinuria that had progressed since birth. He had previously been diagnosed with mildly symptomatic cCMV infection (hepatosplenomegaly, hepatitis, and thrombocytopenia and a positive urine culture for CMV) at birth. An open renal biopsy showed both increased the mesangial matrix and glomerular sclerosis consistent with genetic CNS and nuclear and cytoplasmic CMV inclusions in the tubular epithelia. Genetic testing and long-term follow-up were not reported.

In 1993, Mena et al. [40] reported five consecutive infants with cCMV infections that developed increased urinary output, hypernatremia, and hyposthenuria. The former did not respond to a water-deprivation test, as would be expected in nephrogenic diabetes insipidus (DI), but, instead, normalized after the administration of subcutaneous vasopressin. Therefore, a diagnosis of central DI was made, and a treatment with intranasal desmopressin acetate was initiated, with a resolution of the symptoms in all but one of the infants. The cortisol levels and thyroid function studies were normal in all the cases. Interestingly, all the patients were affected with severely symptomatic cCMV infections, including microcephaly, brain calcifications, and chorioretinitis, but the magnetic resonance imaging showed normal-appearing infantile pituitary glands and hypothalamus. To the best of our knowledge, no further cases of central DI associated with cCMV infection have been reported since then.

### 3.3. Kidney Involvement at the Radiological Level

In a cross-sectional study, Picone et al. [48] described the US findings in 69 pregnancies with cCMV infections, mostly caused by maternal primary infection in a 10-year period in France. US abnormalities were found in 30 out of 69 cases (43.5%), and the central nervous system featured in 26. The most common US findings were hyperechogenic bowels, intrauterine growth restriction, microcephaly, brain calcifications, and ventriculomegaly. Hyperechogenic kidneys were reported in three fetuses (4.3%), all of whom were also affected with severe central nervous system CMV-associated disease. Previously, Choong et al. [39] reported an infant with symptomatic cCMV infection (intrauterine growth restriction, microcephaly, enlarged ventricles, echogenic foci in the basal ganglia at birth, deafness, failure to thrive, and developmental delay at 6 months of age), in whom symptom-free hyperechogenic large kidneys were detected during the third trimester of pregnancy and at birth, which the authors attributed to “renal inflammation resulting from CMV infection”. In an Italian case series [31], hydronephrosis was described in only one out 154 (0.6%) fetuses with cCMV infections; the infant was later diagnosed with symptom-free cCMV infection. Of note, several series describing US findings in fetuses with cCMV infections were excluded from this systematic review, because they did not report kidney abnormalities at all (*n* = 73 [60] and *n* = 237 [61]).

## 4. Discussion

In cCMV infections, the virus is transmitted hematogenously through the placenta and infects many organs in the fetus, as demonstrated by the wide range of clinical findings and laboratory abnormalities that have been described in severely affected cCMV-infected neonates. The kidney is one of the target organs of CMV infection [1]. The virus infects the epithelium of the renal tubules, where it replicates; urine shedding of viral particles is common after primary CMV infection in different populations for long periods of time [2,62]. Actually, children with cCMV infections exhibit the highest prevalence of urine CMV shedding and duration of shedding (up to 5 years of age) when compared with postnatal-infected healthy children or adults diagnosed with sexually transmitted diseases [2], and urine molecular tests have become the gold standard for the diagnosis of cCMV infections [22]. In spite of this, CMV kidney disease has scarcely been described in infants with cCMV infections but, rather, has been reported in older children and adults with cellular immunodeficiencies [6,7,8,11]. To the best of our knowledge, the present systematic review is the first to summarize the available evidence on renal involvement in cCMV infections.

Our systematic review identified only 24 articles and 100 patients with cCMV infections in whom renal involvement was described. The level of evidence of the articles that were included was classified as very low. Approximately half of the articles were more than 20 years old. Most data were gathered from case series and case reports, known to be associated with a high risk of selection and publication bias. Moreover, the data were very heterogeneous, and comprehensive descriptions of the full pathologic, functional, and clinical pictures of infants with cCMV infections and kidney involvement were reported for only one patient [51], limiting the interpretation of the findings and the possibility to infer causality between the viral infection and the renal disease. Only three studies [28,40,51] reported the use of an antiviral treatment, although scarce details were given. Given the type of studies and the large heterogeneity of the data that were reported, we could only perform a descriptive analysis, and we could neither estimate the effect of any interventions nor calculate other outcomes; this is an intrinsic limitation of systematic reviews based on case reports and case series.

Most of the included articles reported pathological data. The former mainly comprised necropsic studies of fetuses or infants with severe disseminated cCMV infections, which did not report information about renal impairment at the functional or clinical level. As expected, typical CMV inclusion bodies in the nucleus (“owl’s eye” cells) and the cytoplasm affecting the renal tubules were universally described [63]; glomeruli were less commonly and less severely affected. CMV nuclear inclusions characteristically infect the epithelia and are the result of active viral replication [64]. Therefore, it is likely that the kidneys of asymptomatic or mildly symptomatic cCMV infants would show similar pathologic features, given the very high viral loads that were observed in urine samples during the first years of life in these patients [65].

Interestingly, lymphocytic interstitial inflammation and necrosis, to some degree, were also reported by several authors [30,35,37,38,41,44,50]. In kidney transplant recipients, CMV mainly causes tubulointerstitial nephritis, with cytopathic changes in tubular epithelial and endothelial cells; inflammation-free cytopathic changes in glomerular endothelial cells and podocytes have been reported in 30% of patients [66] but very rarely do they cause clinically significant glomerulopathy [9,67]. Disseminated infections are common in severe cCMV infections, and inflammation and necrosis have been demonstrated as well in different tissues [30]. In organs such as the liver, the pancreas, and the lungs, the damage caused by inflammation or necrosis is likely to be partially or fully resolved by parenchymal regeneration. In contrast, in the central nervous system, the former likely leads to permanent lesions and neurologic sequelae [30]. In the kidneys, a transient self-limited interstitial nephritis would explain the different pathologic findings in autopsies and the hyperechogenic kidneys that have seldom been described in the US scans of fetuses known to be CMV-infected, but further studies are needed to confirm this hypothesis. In any case, the pathogenesis of fetal injuries remains uncertain but is probably due to both the direct cytopathic effect of the virus and to the host inflammatory response to CMV-infected cells [68].

In 1975, Stagno et al. [69] reported an association between viruria in cCMV-infected infants and the number and severity of signs or symptoms of cCMV disease. These findings have been replicated since then for amniotic fluid [70,71,72] and for urine [73,74,75] and blood at birth [70,74,76], and their association both with the risk of developing sensorineural hearing loss and other cCMV-associated sequelae have been studied. A positive determination of CMV DNA in the cerebrospinal fluid at birth has also recently been associated with symptoms in the cCMV-infected neonate [77]. The reason why viruria at birth correlates with cCMV disease in other organs but does not seem to cause kidney damage in cCMV-infected infants is intriguing. Prospective studies assessing the renal functions in both symptomatic and asymptomatic cCMV-infected neonates are needed to answer this question.

The cardinal feature of CNS is the extensive leakage of plasma proteins into urine through the glomerular capillary wall by the age of 3 months [78]. CMV infection is claimed to be the most common secondary cause of CNS, although evidence of this causality remains uncertain [79]. CNS is also the renal condition that has most often been associated with both cCMV and postnatal CMV infections in young infants [28,41,51]. This association is puzzling, considering the pathologic findings in renal tissues of cCMV-infected patients (i.e., tubular inclusion bodies) and the pathophysiology of CNS (i.e., a glomerulopathy). Three articles in this review reported CNS in 25 patients with cCMV. Chatterjee et al. [28] reported the largest number of patients and, interestingly, also reported elevated levels of inflammatory markers in these neonates as compared to different well-matched control groups, diffuse mesangial sclerosis in 60% of the patients with CNS, and a response to antiviral therapy in some of them. Unfortunately, comprehensive details on the diagnoses and evolution of CNS were not reported, and genetic studies were not performed. Genetic studies confirmed Finnish-type CNS in three out of the remaining five patients with CNS and cCMV [41,51]. Several articles reporting infants with CNS were excluded after the full-text review due to the diagnosis of CMV infection being made beyond 3 weeks of age [79,80,81,82,83,84,85,86,87,88,89,90,91,92,93]; additionally, four infants in the Chen case series were excluded for the same reason [51]. Overall, 21 patients with concomitant CMV infection (cCMV, *n* = 5; CMV infection diagnosed beyond 3 weeks of age, *n* = 16) and CNS have been reported (Table 5). Among the former, 12 out of 13 patients with kidney biopsy results showed pathological findings consistent with those that have been described in the different genetic forms of CNS [94], and genetic mutations of these diseases were confirmed in five out of 12 cases in whom genetic studies were performed. Additionally, 15 patients received antiviral treatment, which led to some degree of CNS improvement in 10 of them. The temporality and the response to a specific treatment point to a possible causal relationship between CMV infection and CNS.

**Table 5 microorganisms-09-01304-t005:** Details of patients diagnosed with concomitant congenital nephrotic syndrome and CMV infection.

Ref.	Kidney Histopathological Findings	Genetic Study	Response to Antivirals
**PATIENTS WITH CONFIRMED CONGENITAL CMV INFECTION**			
[54]	NR	*NPHS1*	No
	NR	Negative	Yes
	Mesangial proliferative glomerulonephritis	*NPHS1*/*COL4A5*	Partial response
	NR	*NPHS1*	No
[41]	Sclerosing glomerulonephritis and nuclear and cytoplasmic inclusions in the tubular epithelia; inflammatory infiltrate	ND	Not treated
**PATIENTS IN WHOM CMV INFECTION WAS DIAGNOSED BEYOND 3 WEEKS OF AGE**			
[54]	NR	Negative	Yes
	NR	Negative	Not treated
	NR	Negative	Yes
	NR	Negative	Yes
[88]	Occasional sclerosed glomeruli and tubular atrophy	ND	Not treated
[87]	Mesangial proliferative; lack of CMV inclusions	ND	Yes
[80]	Diffuse mesangial sclerosis and CMV inclusion bodies in the tubular cells and in some glomeruli	ND	Yes
[82]	Mesangial proliferative; lack of CMV inclusions	ND	Not treated
[91]	Mesangial proliferative and sclerosing glomerulonephritis	*NPHS2*	No
[89]	Mesangial proliferative and sclerosing glomerulonephritis	ND	Yes
[86]	Mesangial cell hypertrophy with interstitial edema; no diffuse mesangial sclerosis	Not *NPHS1/2* or *WT1*	Yes
[83]	Mesangial proliferative and sclerosing glomerulonephritis, interstitial infiltrates and fibrosis/tubular atrophy	*NPHS1*	No
[81]	Normal tissue	ND	No
[84]	Consistent with Finnish-type CNS	Not *NPHS1/2* or *ACTN4*	NR
[85]	Hyalinization in glomeruli, vacuolization in some tubuloepithelial cells, and scarce interstitial mononuclear cells	ND	NR
[95]	NR	ND	Partial response

Abbreviations: CNS, congenital nephrotic syndrome; ND, not done; NR, not reported.

Nevertheless, considering the high prevalence of CMV infection worldwide [62], a higher incidence of CMV-associated CNS would be expected, and the pathophysiologic mechanism of the disease remains uncertain. Alternatively, CMV infection might only be an accompanying condition rather than the cause of CNS. In fact, a genetic analysis is always recommended to exclude hereditary CNS [78], even when the CMV infection or other secondary causes are detected.

Central DI, potentially due to severe brain disease in five infants with cCMV infection, was reported in 1993 [40]. Brain malformations, tumors, and other infiltrative diseases are the main cause of central DI in the pediatric age [95]. Very rarely, central DI has also been reported with other infections of the central nervous system [96,97,98]. In Mena´s series [40], neuroimaging did not identify abnormalities in the hypothalamus or pituitary glands. The loss of a high signal in the posterior pituitary gland in T1-weighted images has been reported in the majority of cases of central DI [99], but normal magnetic resonance imaging in children with primary central DI has been reported as well [100]. Although mild cases may have gone underdiagnosed, the lack of further reports in the literature calls into question the association between cCMV and central DI.

Likewise, the data show that cCMV infection is not associated with a higher risk of kidney malformations. Hyperechogenic kidneys have rarely been reported, usually in severely cCMV-affected fetuses, but their incidence and pathophysiological significance remain uncertain. Hyperechogenic bowels, a nonspecific ultrasound finding in cCMV, has been attributed to intestinal plexus involvement with reduced neuronal functionality that may lead to peristalsis impairment [101].

Our systematic review has further limitations that need to be acknowledged. With significant renal involvement in infants with cCMV infections probably being a rare event, it is possible that mild cases may have gone underdiagnosed and/or underreported by the authors. A potential publication bias in cases with fatal outcomes can also not be excluded. Some of the articles were published a long time ago, when the microbiological methods that are nowadays considered the gold standard for the diagnosis of cCMV infection [62] were not yet developed (i.e., molecular methods) or were available only in referral labs (i.e., conventional or shell assay viral cultures). Nevertheless, we decided to include those studies where strong cytopathologic or immunohistochemical evidence of CMV infections were demonstrated in kidney tissues, despite the lack of microbiological confirmation [63]. Similarly, we used the currently accepted 3 weeks of age cutoff to define cCMV infection, as opposed to postnatal infection or late cCMV infection diagnoses, which could not be firmly established. The latter may have led to the exclusion of patients with cCMV and CNS, as mentioned above. We also adhered to the classification of symptomatic disease at birth and its severity, as published in 2017 [22]; previous differing definitions on symptomatic or asymptomatic cCMV infections would challenge the interpretation of the literature.

## 5. Conclusions

In summary, this systematic review showed that the virus consistently infects the kidney and actively replicates in the epithelium of tubular cells in children with cCMV. Whether this leads to renal impairment at the functional level remains to be explored. The data also suggested that cCMV infection is not associated with a higher risk of kidney malformations in the fetus. We failed to clarify the role that CMV infection (whether acquired congenitally or postnatally) plays in the pathogenesis of CNS, if any. Nevertheless, clinically significant kidney disease does not seem to be part of the natural history of cCMV. Larger, prospective studies that also include asymptomatic or mildly symptomatic cCMV-infected infants are needed. The implementation of noninvasive markers will be key to improving our understanding of the impact of cCMV in the renal functions of these otherwise mostly healthy infants.

## Figures and Tables

**Figure 1 microorganisms-09-01304-f001:**
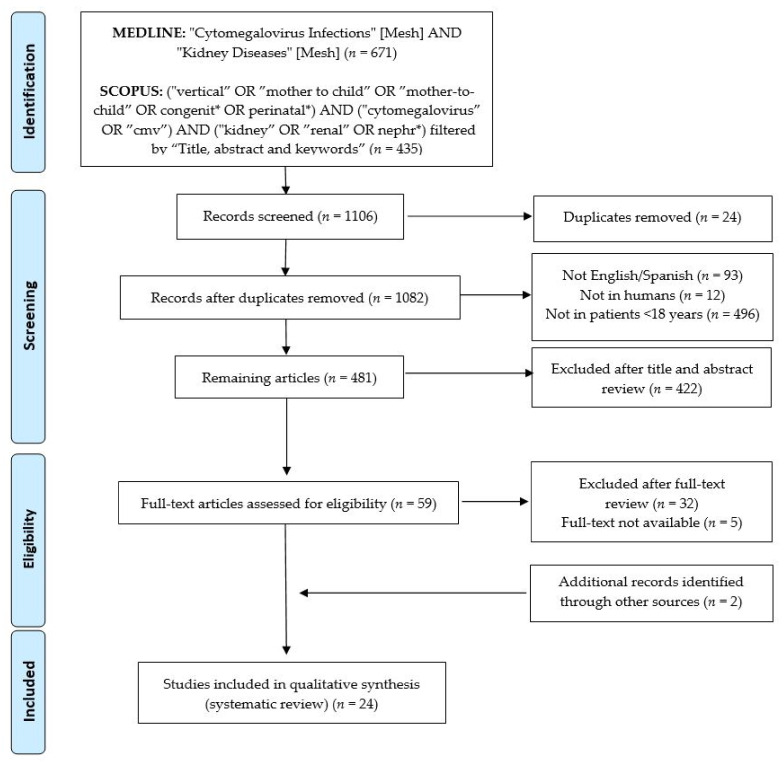
PRISMA flowchart of the current systematic review process.

## Data Availability

Data sharing not applicable.

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
