# Peer review of "Renal Involvement in Congenital Cytomegalovirus Infection: A Systematic Review"

_microorganisms, 2021, doi:10.3390/microorganisms9061304_

Round 1
Reviewer 1 Report
This is a well written, interesting manuscript that will contribute to the literature.
Author Response
We thank the Reviewer for the kind comments about our manuscript. The English language and style have been edited; we hope the paper now reads better.
Reviewer 2 Report
In the present work, Rios-Barnes and colleagues perform a systematic review on the effects of HCMV congenital infection on kiney tissues. HMCV congenital infection is a leading cause of disease worldwide, and with no vaccines or approved antivirals to prevent vertical transmission, it represents a serious threat to human health. After defining search, inclusion and inclusion criteria, authors identify and analyze 22 manuscripts, most of which published more than 20 years ago, when HCMV detection was still in its infancy. Unfortunately, this results in the manuscript not being able to draw any reliable conclusion. Despite the authors clearly state such limitation, the outcome of such work is poorly informative, since only very low level of evidence emerged from their analysis. Therefore the utility of the present manuscript is dubious, and this reviewer is not sure that the available data can fully support the author's conclusions.
Beside this important limitation, the manuscript is clearly organized and written.
However I have several specific concerns that need to be fully addressed should the Editor decide to ask for a revision.
- page numbers are completely messed up so that the manuscripts end with page 15 of 25.
- the authors should explain what they really mean with "necropsy", that to this reviewer means "autopsy on a dead animal". Maybe the term "autopsy should be rather used"
- Similarly, it is not clear what the authors mean with the term "sanctuary" that to the knowledge of this reviewer is usually intended to a tissue or anatomical site where antiviral drugs cannot efficiently penetrate.
- - line 26 "does not associate an increased..." is "with" missing here after "associated"?
- -line 34 "beta Herpesvirinae subfamily" seems more appropriate here.
- -line 44 "latent infection". It is not clear what the authors mean here. Does "latent" imply "clinical latency" or what? If there is persistent replication going on, I doubt this might contribute to latent infection, I strongly recommend rephrasing.
- line 62. The authors should explain the difference between the "prophylactic" and "pre-emptive" strategies here.
- Line 72 "reinfection with a different CMV strain". The authors should provide more information here regarding the kind of immune protection a primary infection elicits in patients, and its ability to protect from reinfection with the same genotype or a different one. Also informations regarding cross-projectivity between different HCMV strains (and genotypes) should be provided, along with relevant references.
- Figure 1. A better resolution image should be provided.
- Table 3: the column heading are not clear, and the meaning of "+++", "++" and "+" should be defined in the Table legends.
- It is not clear if the authors also investigated kidney function/anatomy damage in patients with long-term sequelae.
- Table 5: a legend explaining the meaning of columns headings should be included.
- Line 556 - is there something missing here?
Line 594 ".... whether this leads to renal impairment at the functional levels remains to be explored". Wasn't this the aim of the Review?
Reviewer 3 Report
The review titled “Renal involvement in congenital cytomegalovirus infection: a systematic review” comprehensively summarized the available data and existing literature on renal involvement in congenital cCMV infection. The present article is well-written and well-structured with categorization of case reports and findings. However, author could also include the observations of a case report titled “Isolated renal involvement of cytomegalovirus inclusion disease in an infant” (PMID: 29456230) which showed transient renal failure, renomegaly, and renal biopsy with CMV inclusion bodies without the involvement of other systems. Authors could also discuss about another case report “Of “Cotton Balls” and “Owl’s eyes” (PMID: 33361734) which described a case of congenital cCMV infection along with pneumocystis jiroveci pneumonia from HIV infected parents where kidney showed congested glomerular capillary loops with CMV inclusions in the renal tubular epithelium and glomerular capillaries. I don’t have major concerns except these few minor suggestions.
Author Response
We thank the Reviewer for the kind comments about our manuscript. The English language and style have been edited; we hope the paper now reads better.
With regards to the 2 articles the Reviewer recommends, they both describe patients in whom CMV infection was diagnosed beyond the age of 3 weeks and, therefore, did not fulfill our inclusion criteria. This also happened with several cases of congenital nephrotic syndrome together with CMV infection in young infants, as summarized in the Discussion and Table 5.
Round 2
Reviewer 2 Report
I think the authors have carefully addressed all my minor comments, although I am still not convinced the manuscritpt deserves publication. Indeed, my main concern, i.e. the lack of adequate studies on the topic, still stands and can not be addressed by a revision.